

# How to understand limitations of generative networks

Ranit Das[1], Luigi Favaro[2], Theo Heimel[2],
Claudius Krause[2], Tilman Plehn[2] and David Shih[1]

**1** NHETC, Department of Physics & Astronomy, Rutgers University, Piscataway, NJ USA
**2** Institut für Theoretische Physik, Universität Heidelberg, Germany

## Abstract

Well-trained classifiers and their complete weight distributions provide us with a well-motivated and practicable method to test generative networks in particle physics. We illustrate their benefits for distribution-shifted jets, calorimeter showers, and reconstruction-level events. In all cases, the classifier weights make for a powerful test of the generative network, identify potential problems in the density estimation, relate them to the underlying physics, and tie in with a comprehensive precision and uncertainty treatment for generative networks.



# 1 Introduction

Like all of society, LHC physics is currently undergoing a transformation driven by modern data science. The experimental and theoretical methods of LHC physics have always been numerical in nature, with the goal to quantitatively, systematically, and comprehensively understand data in terms of fundamental theory. Generative networks are an exciting concept of modern machine learning (ML), combining unsupervised density estimation in an interpretable phase space with fast and flexible sampling and simulations [1]. Currently, the most promising architectures for precision generation are normalizing flows and their invertible network (INN) variants, but we will see that diffusion models and generative transformers might offer an even better balance of precision and expressivity.

The range of tasks for generative networks in LHC simulations and analysis is extensive. Given the modular structure of LHC simulations, it starts with phase space integration and sampling [2–7], for instance of ML-encoded transition amplitudes. More LHC-specific tasks include event subtraction [8], event unweighting [9, 10], or super-resolution enhancement [11, 12]. Generative networks working on physics phase spaces have been developed and tested as event generators [13–18], parton showers [19–23], and detector simulations [24–48]. These networks should be trained on first-principle simulations, easy to handle, efficient to ship, powerful in amplifying the training samples [49, 50], and — most importantly — precise. Going beyond forward generation, conditional generative networks can also be applied to probabilistic unfolding [51–56], inference [57, 58], or anomaly detection [59–64], reinforcing the precision requirements.

For all the above tasks, normalizing flows or INNs have reached the level of precision, stability, and control required by LHC physics. Methods to control the performance of these generative networks include Bayesian network setups [18, 65], classifier-reweighting [18, 66–68], and conditional training on augmented data [18]. Building on these developments, LHC physics needs methods to systematically evaluate the performance and the precision of generative networks [69], for example to quantify possible gains through new architectures [39, 70–72].

In this paper we will explore the merits of a classifier-based evaluation of generative networks in particle physics. We will start by defining the goals of such a systematic evaluation and then introduce the classifier metric in Sec. 2. We will present our approach for jet generators [69] in Sec. 3, and discuss it in more details for a calorimeter simulation similar to Ref. [33] in Sec. 4. Finally, we will show how to use event weights to track progress between two versions of an ML-event generator [18] in Sec. 5. We will also illustrate how a systematic scan over kinematic distributions of events with anomalous weights can identify issues of a trained network and how Bayesian networks help us identify the reason for this discrepancy.

All three applications combined illustrate how the distribution of learned control weights over phase space is a reliable measure of the quality if the generative networks and that its shape provides a powerful "explainable AI" (xAI) tool which allows us to systematically search for failure modes of generative models, identify the underlying physics cause, and improve the tested networks efficiently.

# 2 Testing generative networks

Given a generative model trained on some reference data, we would like to know how well it reproduces the data in the full phase space. This includes correct reproduction of critical high-level features, such as transverse momenta and invariant masses in the case of event generation, or shower profiles and MIP peaks in the case of calorimeter simulation. But it also includes all the multi-dimensional correlations between all the features throughout phase space, which might not be visible at the level of histograms of pre-defined high-level features.

We know some typical failure modes of generative networks [18], including features completely removed by a fit-like density estimation, washed-out features with poor resolution, underpopulated kinematic tails, or wrongly learned phase space boundaries. Comparing kinematic distributions of generated and training events allows us to identify many of these issues, making use of the fact that phase space is interpretable and we can typically derive phase space distributions using first principles in quantum field theory or detector design. However, looking at pre-defined phase space distributions runs the risk that we miss a problem, for example when it only affects complex correlations.

Clearly, sensitive metrics are needed to assess the quality of a generative model throughout all of phase space. These metrics should be both multi-variate (capturing all correlations) and interpretable (offering a way to diagnose which critical high-level features are most discrepant). Ideally, these metrics could also offer a systematic way of improving the generative model.

An optimal binary classifier, trained to distinguish generated from reference data in the full phase space, fits the bill in every respect. By the Neyman-Pearson (NP) lemma, this classifier is the most powerful discriminant between generative model and reference data. It is already well-established that one can use the classifier to *reweight* the generative model and bring it closer to the reference data [66]. However, simply reweighting the generated data can lead to problems, for instance in regions with large weights. This can be remedied by using the classifier weights to improve the generative model [18,67,68]. This is also the basis of training GANs, where the training objective is to minimize the difference of the generated distribution to the reference distribution quantified by a classifier network. However, in this case the classifier is not trained to reach close to optimal discrimination power to allow for an equilibrium between generator and discriminator. In addition to these methods to directly improve the generative model with the classifier weights, they can also be used as a diagnostic tool, which is the focus of this work. By examining the generated and reference data as a function of the cut on the classifier, one can zoom in on the most anomalous regions of phase space, *i.e.* those that are worst-reproduced by the generative model. This facilitates the interpretability of the classifier metric, which could be further enhanced using recent xAI techniques developed in HEP such as Refs. [73,74].

Studies that have used the classifier metric to judge the quality of generative models have tended to focus exclusively on single numbers [33,35,39,41,42,48,69], like the AUC, the loss, or the accuracy of the classifier. While these aggregate measures certainly have their uses, there is much more useful information to be gleaned from the classifier than a single number [18,43, 75]. For example, a global integral measure such as the AUC will not detect discrepancies in tails of distributions. Also, the AUC becomes less and less informative the closer the generated and reference samples become. Finally, declaring the model with the highest AUC as the "best" model is oversimplistic, because the definition of the "best" generative network depends on what we actually require from the generative network and how we want to use its output.

In this work, we will explore what the distribution of classifier outputs tells us about the quality of the generative model. We will choose to work in terms of *weights* which can be obtained from the classifier outputs $C$ as

$$w(x) = \frac{p_{\text{data}}(x)}{p_{\text{model}}(x)} = \frac{C(x)}{1 - C(x)}, \quad \text{with} \quad C(x) = \frac{p_{\text{data}}(x)}{p_{\text{data}}(x) + p_{\text{model}}(x)}. \quad (1)$$

The assumption is that the NP-classifier learns the density ratio. For a good generative model and an optimal classifier, the weight distribution will typically peak near one, with tails to the left ($w \ll 1$) and right ($w \gg 1$), corresponding to regions of phase space where the generative model is overproducing and underproducing the reference data, respectively. On general grounds, the NP classifier should have an excess of generated events as a small-weights tail of

the distribution, and an excess of reference events as a large-weight tail. Indeed this is a general pattern we will observe in the different examples we consider in this work. Having it the other way around, an excess of true events on the left tail and an excess of generated events on the right, would generate a ROC curve below the diagonal, indicating an anti-classifier. A renaming of the classes would then solve the problem in principle by switching the weights of true and generated events. However, finding an anti-classifier after training would lead to a troubleshooting and retraining of the classifier in practice.

Since phase space is interpretable, we can study patterns and clustering of anomalous weights to learn more about the generative network. For instance, a positive feature or tail missed by the generative training will be resurrected through large weights $w(x_i) \gg 1$, clustered in phase space. A wrongly modelled phase space boundary will lead to small weights $w(x_i) \ll 1$ or even $w(x_i) = 0$, also clustered in phase space.

Local features in the weight distribution, not necessarily along the tails, also carry useful information about the performance of the generative model. A simple example is the smearing of a peak in phase space, at $x_{max}$, corrected by universal weights $w > 1$ around the peak. If the smeared phase space feature dominates the total rate, a maximum in the weight distribution appears at

$$w(x) \approx \frac{p_{data}(x_{max})}{p_{model}(x_{max})} > 1 \,. \tag{2}$$

Depending on the exact shape in the training data and the kind of smearing, the weights enhancing the tails of the smeared peak can, but do not have to produce a second maximum in the weight distribution. We will discuss all of these patterns in the following sections.

The practical reason why we can measure the performance of generative networks with classifiers is the typical precision of the two networks. For LHC events with a relatively small number of particles in the final state, we know that generative networks reach a precision around

$$\frac{p_{data}(x)}{p_{model}(x)} - 1 \sim \begin{cases} 1\%, & \text{INN } [18], \\ 10\%, & \text{GAN } [16]. \end{cases} \tag{3}$$

Classifiers are not fundamentally different from regression networks, so we expect them to learn the density ratio at the sub-percent level [75,76],

$$w(x) - \frac{p_{data}}{p_{model}}(x) \sim 0.1\% \,. \tag{4}$$

Thus it should be possible to obtain classifiers which are precise enough to be sensitive to the failure modes of the generative model. This is consistent with the observation that, while classification with generative models can be useful for small training data sets with low complexity, it is typically outperformed by discriminative models for higher-dimensional data with large training statistics [77–81].

Because the weights are constructed from a classifier, we can use standard methods, such as calibration curves, to ensure the classifier is trained properly. We can also reweight generated samples with the learned classifier and see if they become closer to the reference data; this will be another sign that the classifier approximates well the likelihood ratio.

In our study of the weight distributions for generative models, we can draw inspiration from a similar approach to supervised amplitude regression [75,76,82,83]. There, the weights can be constructed directly from the regression task, because the "generated" amplitude is learned directly from the known theoretical calculation. There, as here, tails of the weight distribution will be induced by stochastic training data, a lack of expressivity of the network, or overtraining [75]. For a well-motivated statistical test we can use the fact that many networks are trained on likelihood losses. Those losses include an uncertainty estimate $\sigma_i$, for instance

Table 1: Hyperparameters of the classifier network applied to the calorimeter simulation and event generation datasets.

| Parameter | Calorimeter | Events $Z + \{1, 2, 3\}$ jets |
|---|---|---|
| Optimizer | Adam | Adam |
| Learning rate | 0.001 | 0.001 |
| LR schedule | reduce on plateau | reduce on plateau |
| Decay factor | 0.1 | 0.1 |
| Decay patience (epochs) | 5 | 5 |
| Batch size | 1000 | 1024 |
| Epochs | 150 | 50 |
| Number of layers | 3 | 5 |
| Hidden nodes | 512 | 256 |
| Dropout | 10% | 10% |
| Activation function | leaky ReLU | leaky ReLU |
| Training samples | 60k | 2.7M / 750k / 210k |
| Validation samples | 20k | 300k / 80k / 20k |
| Testing samples | 20k | 3.0M / 830k / 240k |

from a Bayesian regression network, so we can supplement the weight distributions by a pull and analyse both [75],

$$
w_i = \frac{A_{i,\text{data}}}{A_{i,\text{model}}}, \qquad \text{and} \qquad t_i = \frac{A_{i,\text{model}} - A_{i,\text{data}}}{\sigma_i} = \frac{A_{i,\text{model}}}{\sigma_i} (1 - w_i). \tag{5}
$$

The pull should follow a standard Gaussian for uncorrelated stochastic deviations. The combination of weights and pulls it is extremely useful for testing regression networks, so we will try to generalize it to generative networks.

Finally, we make the (obvious) observation that the AUC of a classifier can be extracted from weight samples $w(x_i)$ evaluated on training and generated configurations. As a function of the signal efficiency, the ROC curve is a step-wise, monotonically increasing function. Its integral can be estimated by the sum of bins with width $1/N_{\text{gen}}$, the inverse size of the generated dataset. The height of each rectangle is the fraction of weights in the true, or training dataset, which are larger than a given $w_{\text{cut}}$, normalized to $N_{\text{true}}$. The AUC is then given by the sum,

$$
\text{AUC} = \frac{1}{N_{\text{gen}} N_{\text{true}}} \sum_{w_i \in \text{gen}} |\{w | w \in \text{true and } w > w_i\}|, \tag{6}
$$

where $|\{S\}|$ denotes the cardinality of the set $S$. Therefore, by focusing on the weight distribution of the classifier, we are not missing any information otherwise contained in the AUC.

In this paper we will use three standard LHC applications of generative models to develop a common strategy to quantify the performance of the networks. To test the specific generative networks introduced in the following sections, we use a very generic classifier for Secs. 4 and 5, described in Tab. 1. In Sec. 3 we apply the state-of-the-art in jet classification, PARTICLENET-LITE [84].

## 3 Distribution-shifted jets

As a first application, we consider the JetNet example recently used in Ref. [69] to illustrate different metrics for generative models. They distort jets generated by PYTHIA at the particle level and at the distribution level. In the particle level distortions, each particle in the jet is

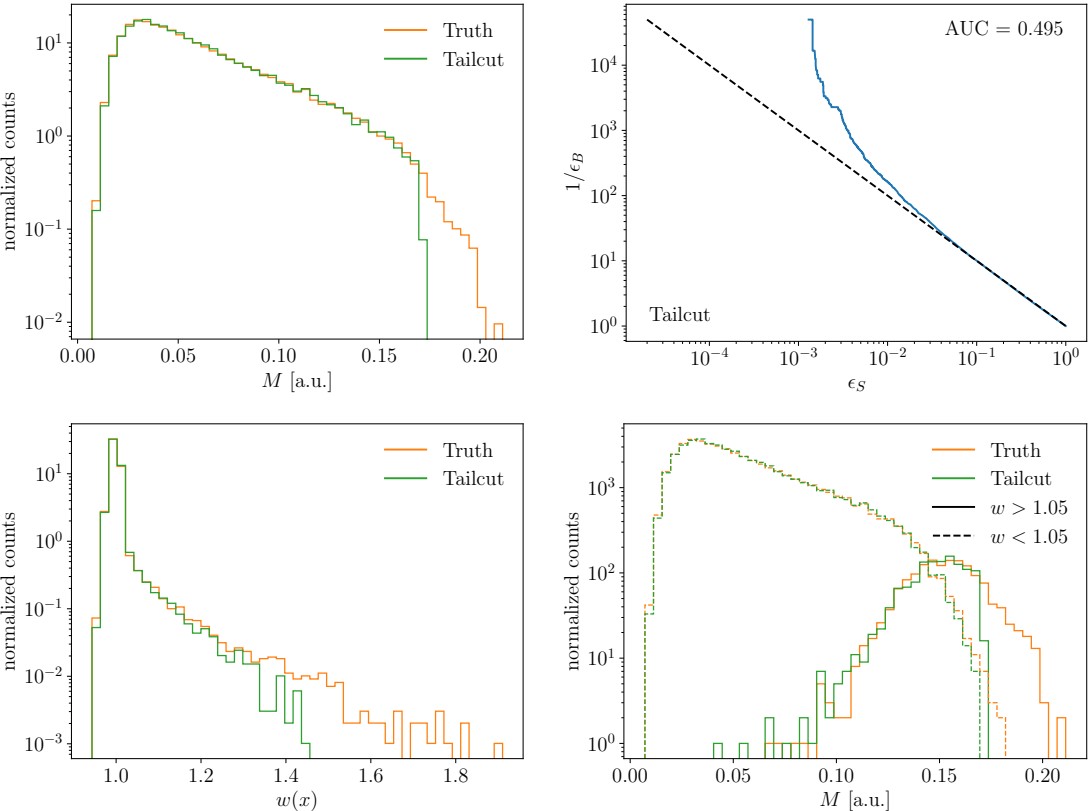

Figure 1: From top left to bottom right: jet mass distribution for the tailcut distortion; ROC curve from the trained ensemble of classifiers; learned weight distribution; jet mass distribution for jets in different classifier weight ranges, to identify clustering.

altered in some way. While this is a realistic scenario, the amount of distortion in Ref. [69] was taken so large, that it makes the classification task almost trivial. For the distribution-level distortions, a single distribution like the jet mass is modified. Jets are reweighted so that all other features and correlations are identical to the reference data, only the one distribution is modified. This is a highly unrealistic toy scenario, and we would not advertize it as physics-motivated, but it provides an interesting challenge for metrics to detect small differences.

In Ref. [69] it was pointed out that the AUC of a classifier metric trained on distribution-level distortion versus reference data is not very sensitive, and metrics such as FID and MMD can detect the flaw in the generative model more sensitively. In line with the general philosophy outlined in Sec. 2, we argue that the AUC is indeed the wrong metric, and examining the distribution of classifier weights, especially the behavior on the tails, is a much more sensitive probe and does detect all distribution-level distortions introduced into the toy generative models.

We perform three distortions on the jet mass, extracted from the relative polar coordinates provided in the JetNet dataset [69]:

1. "Tail cut": remove the tail with an acceptance cut $M < 0.17$;

2. "Smear": smear the distribution by multiplying with a Gaussian with $\mu = 1.0$ and $\sigma = 0.25$.

3. "Shift": shift the distribution by multiplying with a Gaussian with $\mu = 1.1$ and $\sigma = 0.05$;

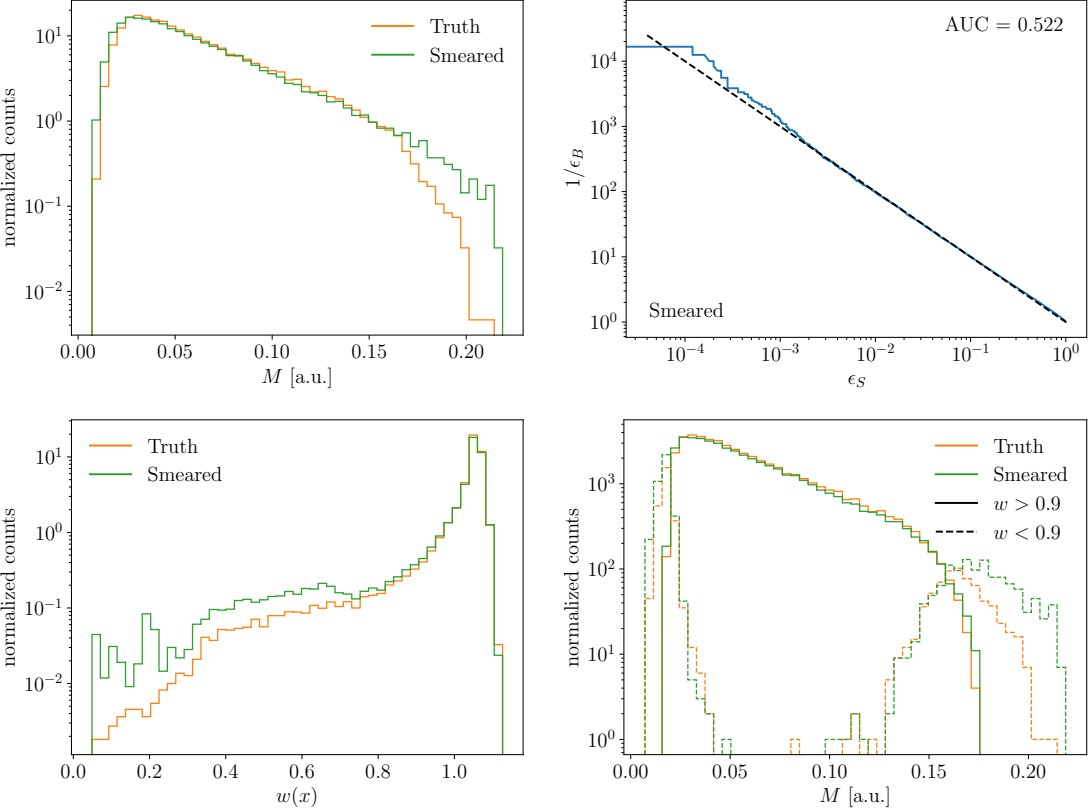

Figure 2: From top left to bottom right: jet mass distribution for the smeared distortion; ROC curve from the trained ensemble of classifiers; learned weight distribution; jet mass distribution for jets in different classifier weight ranges, to identify clustering.

For each distortion, we train the classifier on 100000 distorted and the same number of undistorted jets. The validation set consists of 50000 jets each. In the interest of computation time, we use PARTICLENET-LITE instead of the full PARTICLENET classifier [84] used in Ref. [69]. This only has a minimal effect on the results. For extremely similar datasets and using limited training time we expect a certain variability of the classifier output. To avoid cherry-picking, we combine the five independent trainings of PARTICLENET-LITE, and for from each training select the models with the five lowest validation losses. We ensemble these 25 classifier outputs and verify (by doing it all over again) that this produced a stable, robust result. Evidence that the ensembled classifiers are well-calibrated, and hence learned the likelihood ratio, is provided in Appendix A.

Figure 1 shows the results for the tail-cut case. Ignoring the stochastic nature of the training data, the NP-optimal classifier should be quite singular: all jets with $m < 0.17$ should have classifier weights given by a delta function at $w = 1 - \epsilon$, where $\epsilon$ is the fraction of jets removed from the tail. In addition, there should be a delta function only for reference jets at $w = \infty$. A realistic classifier will transform these two features as a smooth weight distribution. In Fig. 1 we see the smooth weight distribution from our classifier, with nearly all jets populating a sharp peak near one, and a long tail extending to larger values of $w$, solely in the reference sample. This is an example for a large-weight tail expected for a generative model that is missing a tail or a feature. Since the small number of tail jets are soaked up by the bulk, the weight distribution barely changes and only a tail at large weights appears.

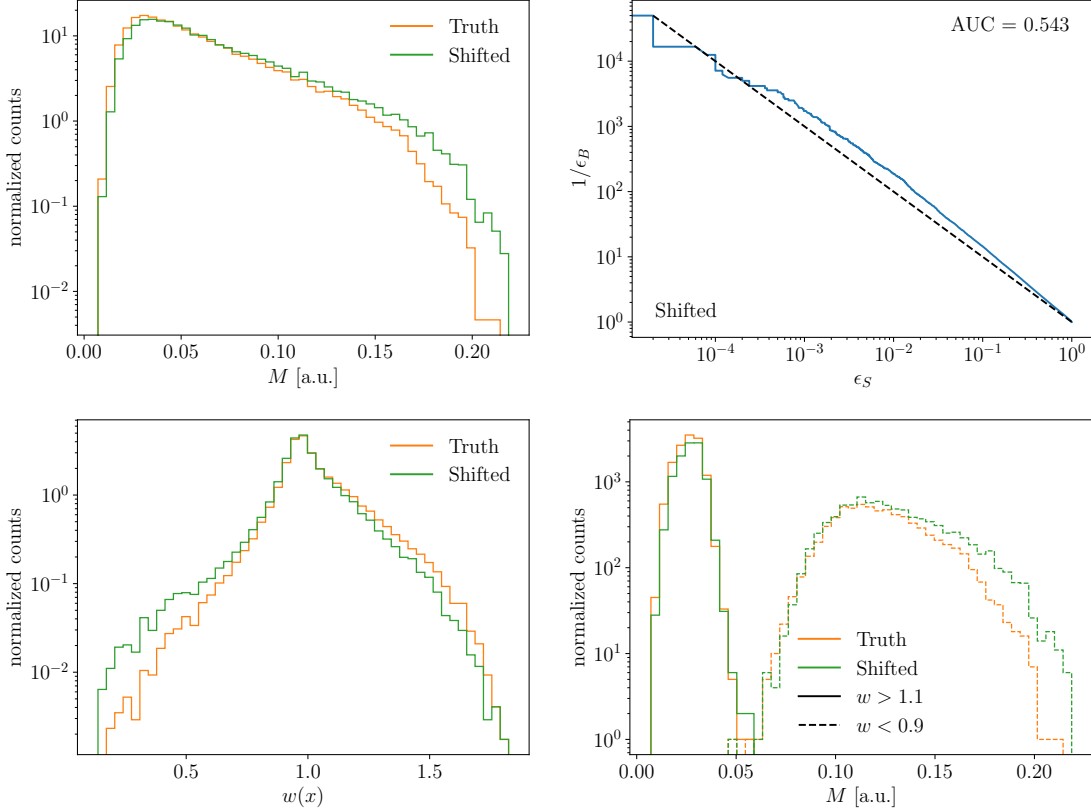

Figure 3: From top left to bottom right: jet mass distribution for the shifted distortion; ROC curve from the trained ensemble of classifiers; learned weight distribution; jet mass distribution for jets in different classifier weight ranges, to identify clustering.

We also see that for the tail cut distortion the AUC is close to 0.5, while even in the ROC curve itself, it is visible that the classifier is far from a fully confused classifier. This confirms that the AUC is a terrible metric for the quality of the generative model. However, the tail of the weight distribution gives us all relevant information. Cutting on the tail of the weight distribution, we correctly identify the discrepancy in the tail of the jet mass distribution. In Sec. 5 we will see how we can even use these weights to recover such a missing feature for a quantitative analysis. This example illustrates nicely how the classifier gives both a sensitive metric of generative model quality and enables interpretability by allowing us to identify in which physics aspect the generative model is wrong.

In Fig. 2, we show the weight distribution for the smeared distortion. The weight distribution has a maximum at $w > 1$ and is dominated by a small-weight tail. This is expected from the general discussion in Sec. 2: the smearing in this case, multiplicative in the jet mass, has a reduced effect at small jet mass and an outsize effect at large mass, and ends up heavily overpopulating the tail of the large-mass regime. Correspondingly, there is hardly any tail with large weights and a very large tail with small weights. Cutting on the small-weight tail correctly reveals the excess of generated jets, now appearing on both ends of the jet mass distribution.

Finally, in Fig. 3, we see the weight distribution for the shifted distortion, again leading to an unhelpful ROC curve and an AUC close to 0.5. Since the distortion is small enough to not significantly overpopulate or underpopulate the tails of the jet mass distribution, the effect on weight distribution is mild and symmetric. We also see the characteristic tilted weight dis-

tribution that indicates a well-calibrated classifier, with generated jets above (below) training jets on the small-weight (large-weight) side. Cutting on the two tails of the weight distribution correctly reveals that the over-population and under-population of generated jets come from the low and high ends of the jet mass distribution, respectively.

# 4  Calorimeter simulation

As a second example of how to use weights over phase space to understand the performance of a generative model, we turn to the classic calorimeter simulation [24, 26, 33, 35], but with a slightly modified INN architecture [85]. We study weight distributions for positron, photon, and pion showers in a simplified calorimeter. The classifier defined in Tab. 1 is trained on voxels, energy, and layer energies in unnormalized shower data. We focus on the classifier with unnormalized preprocessing in this work because it appears to be better calibrated and shows less propensity for overfitting. For more discussion, see Appendix A. As a more realistic scenario, learned calorimeter showers allow us to discussion some aspects of learned weight distributions in more detail.

## 4.1  Tails of weights

In Fig. 4 we show ROC curves and weight distributions for $e^+$, $\gamma$, and $\pi^+$ showers. The top row confirms that positron and photon showers are easier to generate than pion showers. The question is which potential failures are related to this performance difference.

In the second row we show the weight distributions. First, we observe that they are not symmetric, because the reweighting now compensates features. The limit $w(x) = 0$, most visible for the pion shower, marks phase space points where the generator has learned a finite density $p_{\text{model}}(x)$, where the correct density is $p_{\text{data}}(x) = 0$, one of the typical failure mode of generative models discussed in Sec. 2. We will see this more clearly for LHC events in the next section, but mention here that it is not catastrophic if we can enforce corresponding phase space boundaries during generation.

In the third row of Fig. 4 we show the same curves on a logarithmic scale to see the tails. As expected, they are different when evaluated on GEANT and generated showers. Already for positrons, the generated data includes many more showers with $w(x) \ll 1$ than the training data. These are showers for which the generator overpopulates phase space, so they appear preferably in the generated dataset. This tail connects to showers with weight zero.

In contrast, showers with $w(x) \gg 1$ appear more frequently in training dataset. These under-populated regions of phase space correspond, for instance, to features or tails which the network does not learn. This serious failure mode can be identified by evaluating showers with anomalous weights on the training data.

## 4.2  Phase space clustering

The simpler structure of photon showers allows for a detailed study of the clustered observables. By cutting on the weight values and looking at the distribution of the remaining photon showers, we identify three characteristic failure modes highlighted with different colors Fig. 5.

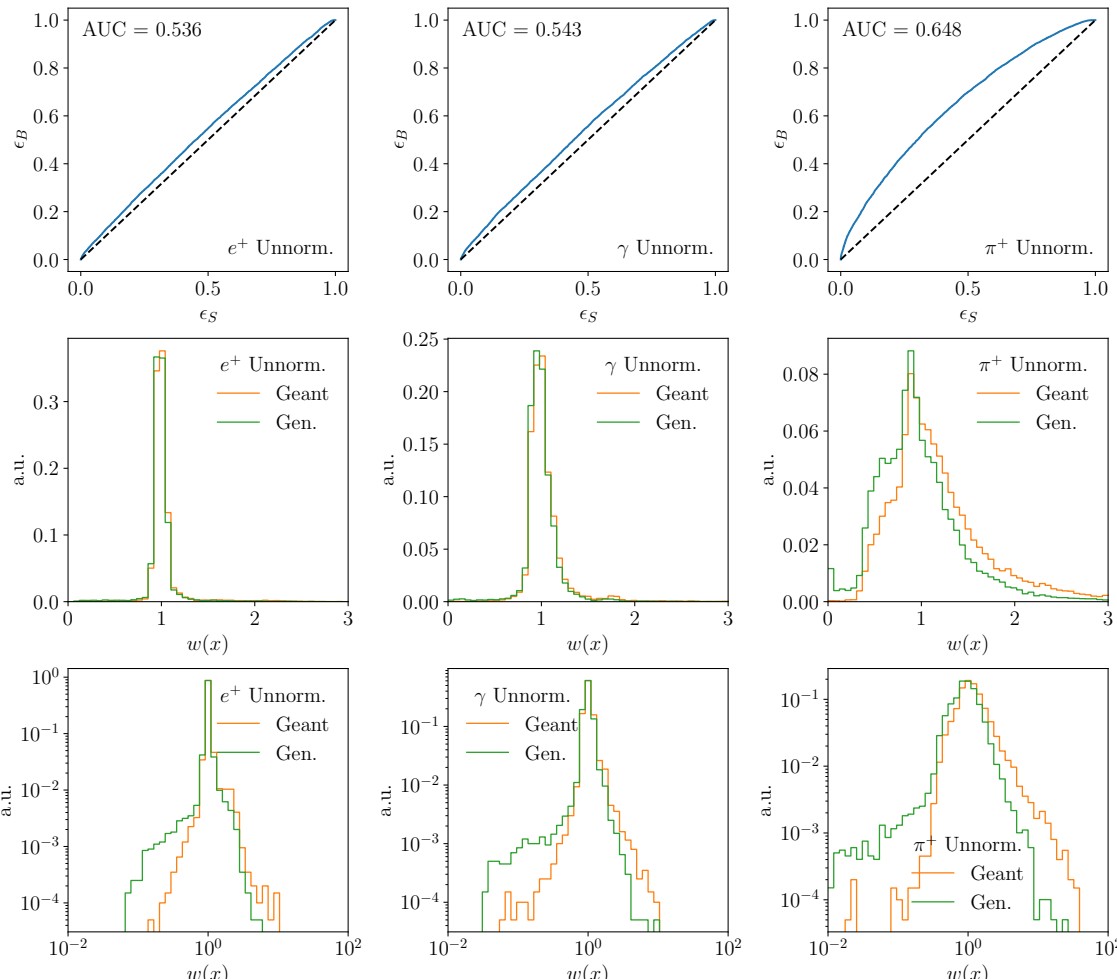

Figure 4: Left to right: calorimeter showers for $e^+$, $\gamma$, and $\pi^+$. Top to bottom: ROC curve, weight distribution on a linear scale, and weight distribution on a logarithmic scale. The weights are evaluated separately on the GEANT dataset used for generator training and the generated dataset.

1. In orange, we isolate the large-weights tail with $w > 1.6$ and no energy deposited in layer 2 ($E_2 < 0.1\,\text{MeV}$), as shown in Figs. 5(c) and 5(f). As shown in Figs. 5(g) and 5(h), these showers have higher sparsity[1] in layers 0 and 1 than the typical shower. Additionally they have lower energy, shown by the $E_1$ histogram in Fig. 5(e), since on average most of the energy is deposited in layer 1. Overall, these showers consist of just a few activated, low-energy voxels in layers 0 and 1, and exactly none in layer 2. This sub-population of showers exists in the GEANT data, but it is not sufficiently generated by the network.

2. In blue, we isolate the small-weights tail with $w < 0.6$. Fig. 5(c) shows that this failure mode is characterized by a single voxel carrying all the energy in layer 2, and Fig. 5(e) shows that this energy is lower than the average energy deposition. Blue and orange agree in every feature that we looked at in layers 0 and 1; they only differ in layer 2. Since these are showers overproduced by the generator, we interpret this as the compensation

---

[1]Here, we are redefining sparsity compared to previous literature [26, 33]: sparsity(here)=1-sparsity(there). This way, higher sparsity means more sparse showers (i.e. showers with only a few voxels activated) while lower sparsity indicates less sparse showers (i.e. showers with many voxels activated).

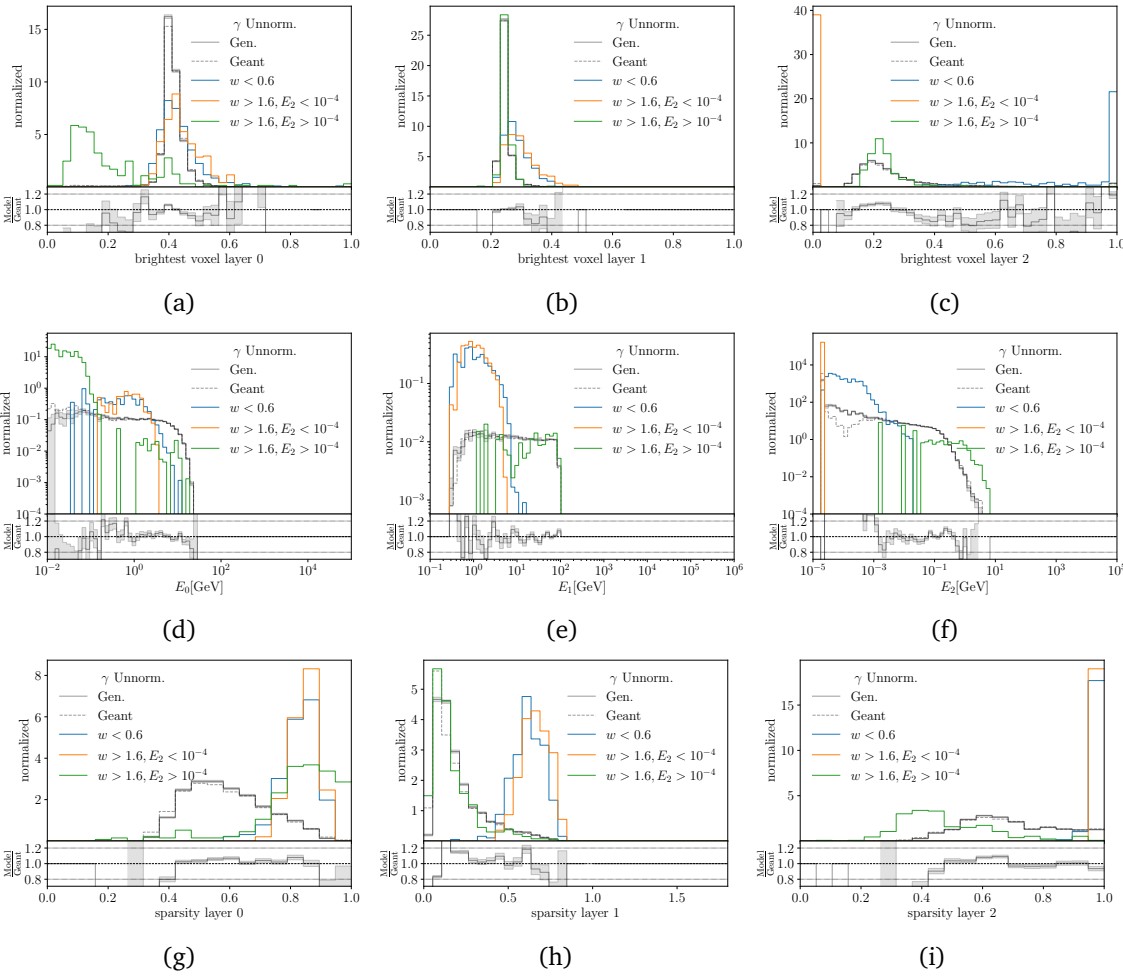

Figure 5: Relevant distributions for $\gamma$ showers in the small-weights (blue) and large-weights regions (orange and green). We show the energy depositions, the fraction of the energy deposited in the leading voxel, and the sparsity in the three layers of the calorimeter.

of the generator for the underproduction of the orange showers; the compensation is only needed in layer 2. We think the reason for both the orange and blue failure modes is due to the low energy and the large number of zero voxels in these showers: this causes them to be especially sensitive to the noise we add during training, since a single voxel is being activated and it either falls just under or just over the minimum energy threshold. The vicinity of these showers to the noise threshold makes it harder for the generator to perfectly model this region of phase space.

3. Finally, in green we isolate again the large-weights tail with $w > 1.6$ that *does* deposit energy in layer 2 ($E_2 > 0.1\,\text{MeV}$). These showers are also underproduced by the generator but they are distinct from the previous two classes. According to Fig. 5(d)-(f), these have very low energy in layer 0 (even lower energy than the orange showers), and higher-than-typical energy in layer 2. In layer 1 their energy is closer to the typical shower. We also see in the sparsity that these photons deposit very little energy and activity in layer 0, while in layers 1 and 2 they are fairly typical. These are showers which develop late in the calorimeter, leaving little or no energy in layer 0. Interestingly, physics tells us that these late-developing showers are possible for photons but

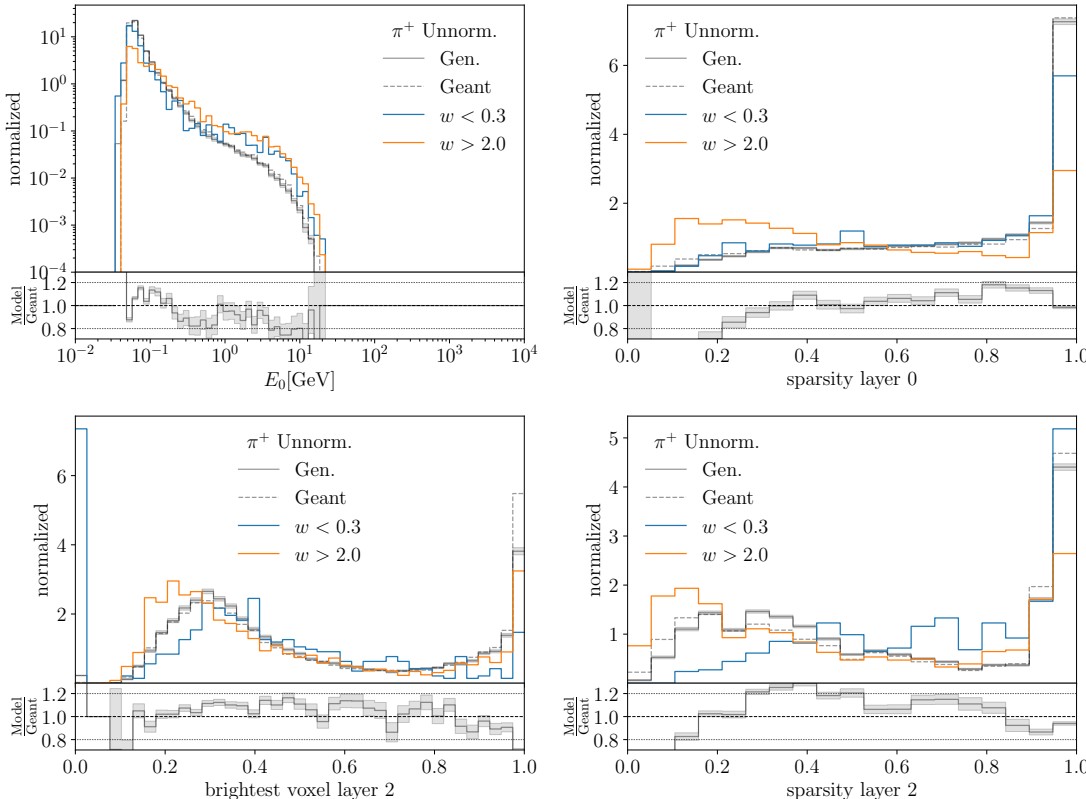

Figure 6: Relevant distributions for $\pi^+$ showers in the small-weights (blue) and large-weights regions (orange). We show the energy deposition and the sparsity in layer 0, and the brightest voxel energy and the sparsity in layer 2.

not likely for positrons. At high energies, the latter interact continuously with the material through Bremsstrahlung, while the former need to convert to $e^+e^-$ first [86]. This leads to showers fully absorbed deeper in the calorimeter, therefore with more energy deposited in layer 2. We see this difference in the physics clearly reflected comparing with the green showers for the positron case (see App.B). The positrons have energy deposited in layer 0, unlike the photons.

The situation becomes much more complicated when looking at pions, where the more complex physics through the nuclear interaction and the poorer generative model make it harder to identify failure modes with kinematic or physics features. In line with the sobering AUC value given in Fig. 4, we see in Fig. 6 that the generator requires correction weights essentially all over phase space. The first distinctive failure mode is corrected by small weights in the energy distributions, for instance in layer 0, which suppress the generated showers to reproduce the sharp lower edge of the energy deposition. In addition, the network produces too many showers with exactly zero energy deposition in layers 1 and 2 (see App.B). They are included in an overflow bin in the energy histograms, but appear as a failure mode in the energy fraction of the brightest voxel, for example in layer 2. Finally, we see showers with large weights cluster at low sparsities. Here the generator has a systematic bias towards simpler showers with fewer voxels. The full set of studied observables for $e^+$, $\gamma$, and $\pi+$ can be found in Appendix B. Given these observations, the leading improvement to the generative model concerns the low-energetic voxels. As discussed before, this can be linked to the addition of noise during training and provides us a research direction to improve the generator, e.g. sampling from a different noise distribution or the development of a noise-less training scheme.

# 5 Event generation

The third network we analyze using learned classifier weights generates events for the process

$$pp \to (Z \to \mu^+\mu^-) + 1, 2, 3 \text{ jets}, \tag{7}$$

at the reconstruction level, using the precision INN architecture described in detail in Ref. [18]. We first use the published version and then the current state of the art, for which we remove the PCA preprocessing, as it introduces correlations between different jet multiplicities which make the training harder. The convergence of the updated Bayesian version is improved by initializing the standard deviations of the trainable weights with a small value, bringing its performance close to the deterministic version.

As in Ref. [18], we train a classifier on the same observables as the generator. Because the classifier does not have an invertibility constraint, we can add more features as network inputs. For LHC events, the generator will wash out intermediate mass peaks and the $\Delta R$ distribution between jets, so we provide the classifier with

$$\left\{ p_{T,i}, \eta_i, \Delta\phi_{i,i-1}, M_i \right\} \cup \left\{ M_{\mu\mu} \right\} \cup \left\{ \Delta R_{i_1,i_2} \right\} \cup \left\{ \Delta R_{i_2,i_3}, \Delta R_{i_1,i_3} \right\}, \tag{8}$$

where $M_i$ is only present for muons and there is no $\Delta\phi$ for the first particle. In addition, to help the network focus on small $\Delta R$, we take the inverse of this observable and apply a cutoff as a preprocessing step.

## 5.1 Standard generator and mass peak

We start the discussion of potential failure modes of the event generator with the old network setup from Ref. [18], and with the weight distributions shown in Fig. 7. This network encounters difficulties in reproducing the $Z$-peak, where the learned width turns out too large for two and three jets. An example of this is shown in Fig. 8 for $Z + 2$ jets. In the upper sub-panels we show the ratio of generated to truth density as a function of $M_{\mu\mu}$, the most discrepant distribution for this generator. We see a characteristic shape in the density ratio aligned with the $M_{\mu\mu}$ distribution. The ratio shows a dip where the model underpopulates the true distributions due to the smearing and two massive shoulders on either side of the peak, where the smearing cause an overpopulation of generated events relative to truth. The trained classifier compensates this density ratio with values as large as $w \sim 1.5$ on the $M_{\mu\mu}$ resonance and $w = 0.6 \dots 0.8$ on its shoulders.

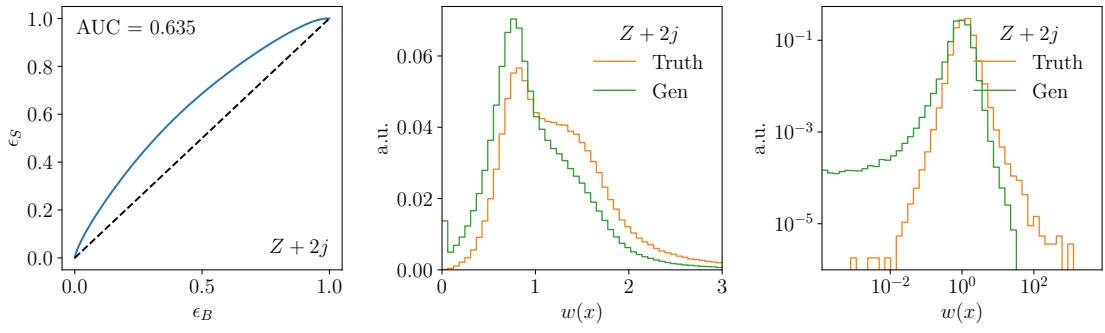

Figure 7: Left to right: ROC curve, weight distribution on a linear scale, and weight distribution on a logarithmic scale for $Z + 2$ jets events, using the outdated standard generator. The weights are evaluated separately on the true, training dataset for the generator and the generated dataset.

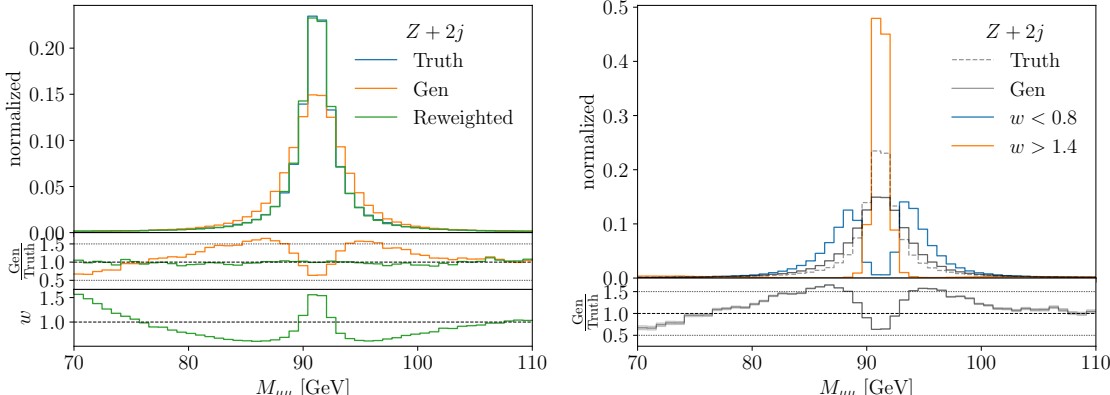

Figure 8: $Z$-peak distributions for $Z + 2$ jets events from the outdated standard generator. We show the agreement between the generated events with the truth or training data (left) and evens in different weight ranges (right). The events with small weights are taken from the generated distribution, the events with large weights are taken from the truth distribution.

The corresponding distribution of trained classifier weights is shown in Fig. 7. In this case, the main peak is shifted to $w < 1$, driven by the overpopulated wings of the smeared $M_{\mu\mu}$ distribution. A secondary peak/shoulder appears around $w \sim 1.5$, corresponding to the underpopulated $M_{\mu\mu}$ resonance. It is interesting to compare this weight distributions from the smeared $M_{\mu\mu}$ resonance and the smeared distortion of the JetNet data in Fig. 2. Although both are driven by a smearing, the weight distributions are very different. For the smeared jets the maximum of the weight distribution appears at $w > 1$, representing the actual peak configurations, while for the LHC events the maximum of the weight distribution is shifted to $w < 1$, driven by the shoulders of the smeared peak. This reflects the clear differences in the form of the smeared phase space feature and the details of the actual smearing. The NP-classifier does not identify the smearing mechanism in the sense of a Wasserstein-distance, but tracks the density ratio over phase space and requires an interpretation of the entire weight distribution and the corresponding interpretable phase space.

## 5.2 State-of-the-art generator and feature scan

Next we turn to an improved version of the $Z$+jets event generator, where the $Z$ mass peak is much improved, and the main failure mode shifts elsewhere. In Fig. 9 we show the same weight distributions as in Fig. 7, but for the updated version of the INN event generator and one to three jets. The central peaks are much more narrow, and the distributions for one and two jets are now almost identical. However, we still observe distinctive tails of the weight distributions. They should be evaluated on generated events, if we are interested in small weights, and on training events, if we are interested in large weights. Even for three jets the maximum of the weight distribution remains at one, indicating that for the updated generator the mass peak is no longer a serious problem. On the other hand, the tail towards large weights is sizeable, indicating that we should look for missing sub-leading features in the generated event sample.

Consequently, we search for phase space clustering of $Z + 3$ jets events with anomalous weights in Fig. 10, similar to Fig. 8. We see the effect of small statistics in the otherwise accurately learned $p_T$-tail, and the $Z$-mass peak with hardly any reweighting required. The angular correlations between the jets is the one distribution that is not described well. While reweighting is not needed to describe the maximum around $R_{jj} \sim 3$, the collinear enhancement

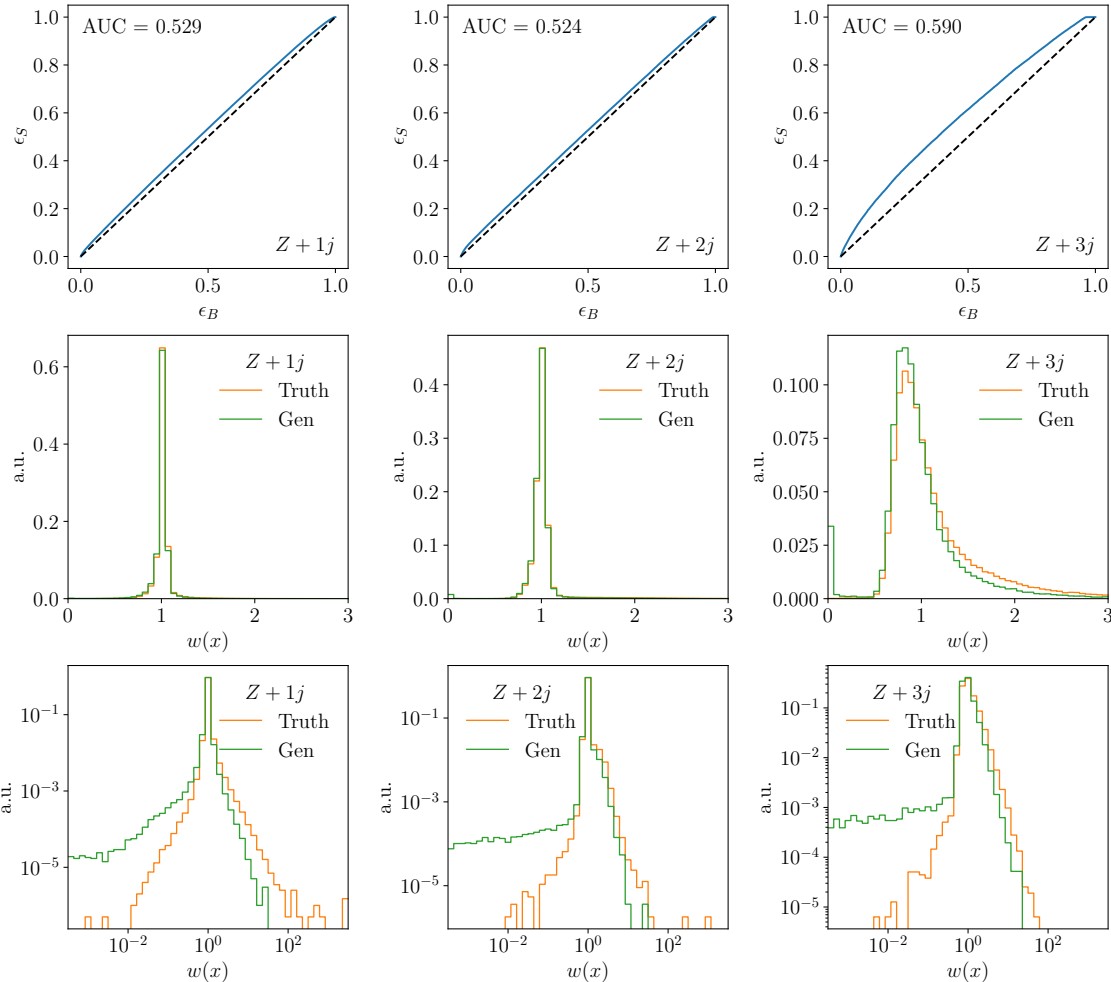

Figure 9: Left to right: $Z + \{1, 2, 3\}$ jets using the state-of-the-art generator. Top to bottom: ROC curve, weight distribution on a linear scale, and weight distribution on a logarithmic scale. The weights are evaluated separately on the true, training dataset for the generator and the generated dataset.

in the range $R_{jj} = 0.4 \dots 1.5$ only appears after reweighting with large weights, while the phase space boundaries for $R_{jj} < 0.4$ requires very small, potentially zero weights. We can confirm this by looking at the events in the leftmost bins in the central row of Fig. 9. These correspond to weights $0 < w < 0.06$, and we have confirmed that for two and three jets at least 95% of these events have one $\Delta R_{jj} < 0.4$.

Finally, we can use event weights to identify unknown issues for a given trained network. In App. B we show a large set of kinematic $Z$+jets correlations for events in the tails of the weight distributions. Two kinematic distribution stick out as poorly described — the rapidity of the softest jet and its jet mass, both shown in Fig. 11. While $\eta_{j_3}$ is part of the standard set of distributions to check, its jet mass is not usually used to benchmark this kind of network [18]. However, it becomes important when we combine event-level with jet-level analysis tools.

In the lower panels of Fig 11 we show the corresponding distributions, to confirm that the reweighted generated events reproduce the truth and the classifier output is correct. The reason for the poor performance on the third jet is, most likely, the small size of the training sample. For a standard, deterministic network the source of such a failure is hard to determine, so we resort to a Bayesian version of the same network for this purpose.

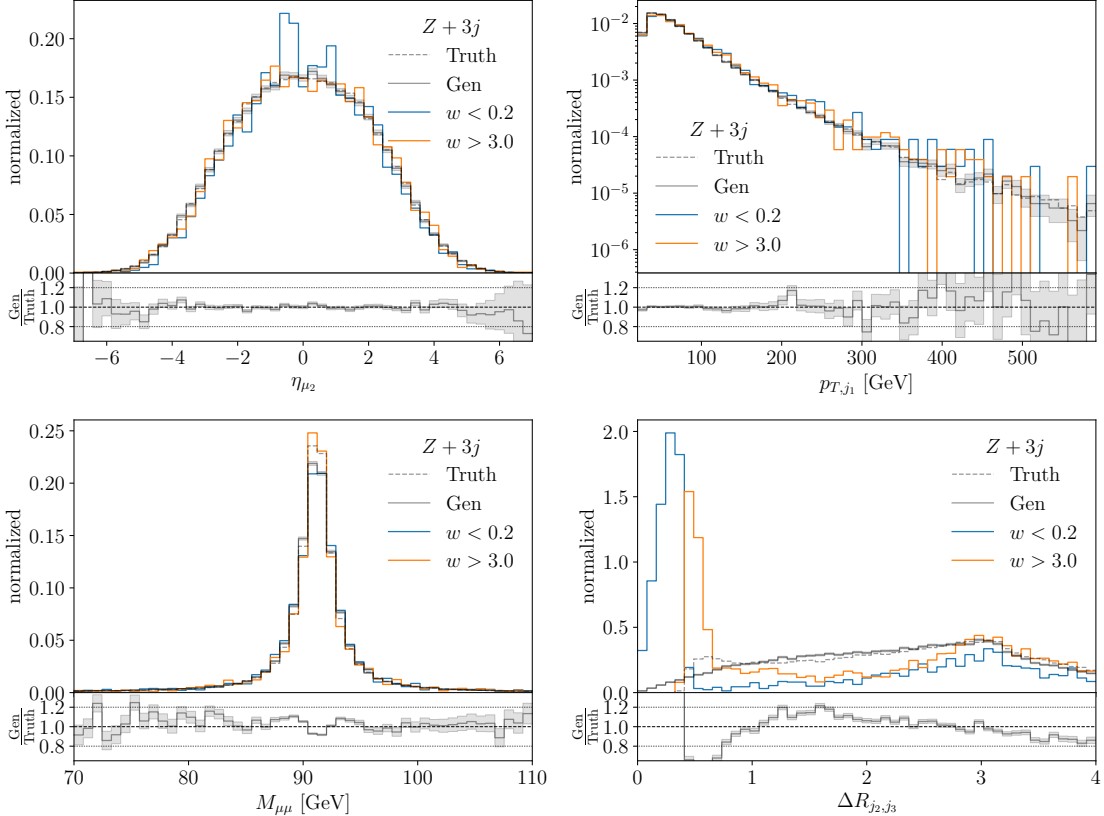

Figure 10: Kinematic distributions for $Z + 3$ jets events from the state-of-the-art generator in different weight ranges, to see if events with large corrections cluster in phase space. The bottom panels show two jet masses, which are not part of the standard requirements testing the $Z$+jets kinematics. The events with small weights are taken from the generated distribution, the events with large weights are taken from the truth distribution.

## 5.3 Bayesian generators and pull

While weight distributions encode a wealth of information about the generative model, we do not know from first principles what shape to expect for well-trained networks. A way out is to supplement it by pull distributions, introduced in Eq.(5), which should approach a standard Gaussian. Deterministic generative networks do not provide us with the necessary information, but a Bayesian generative network returns a density as well as an uncertainty estimate on this density [18,65]. We can then define the mixed ratio

$$t(x_i) = \frac{\mu(x_i)[1 - w(x_i)]}{\sigma(x_i)}, \qquad (9)$$

where the mean of the estimated density $\mu(x_i)$ and its uncertainty $\sigma(x_i)$ are provided by the Bayesian generator, and $w(x_i)$ is the classifier output.

To extract an error on the likelihood for a specific event, we fix the network weights to the maximum of their posterior distribution and generate a dataset. Next, we use the network as a density estimator and extract a distribution of likelihoods for each event by sampling from the network weight distribution. The width of this distribution should give an estimate of the uncertainty. However, we have to be careful in this interpretation, as we cannot treat the event-wise likelihoods as uncorrelated.

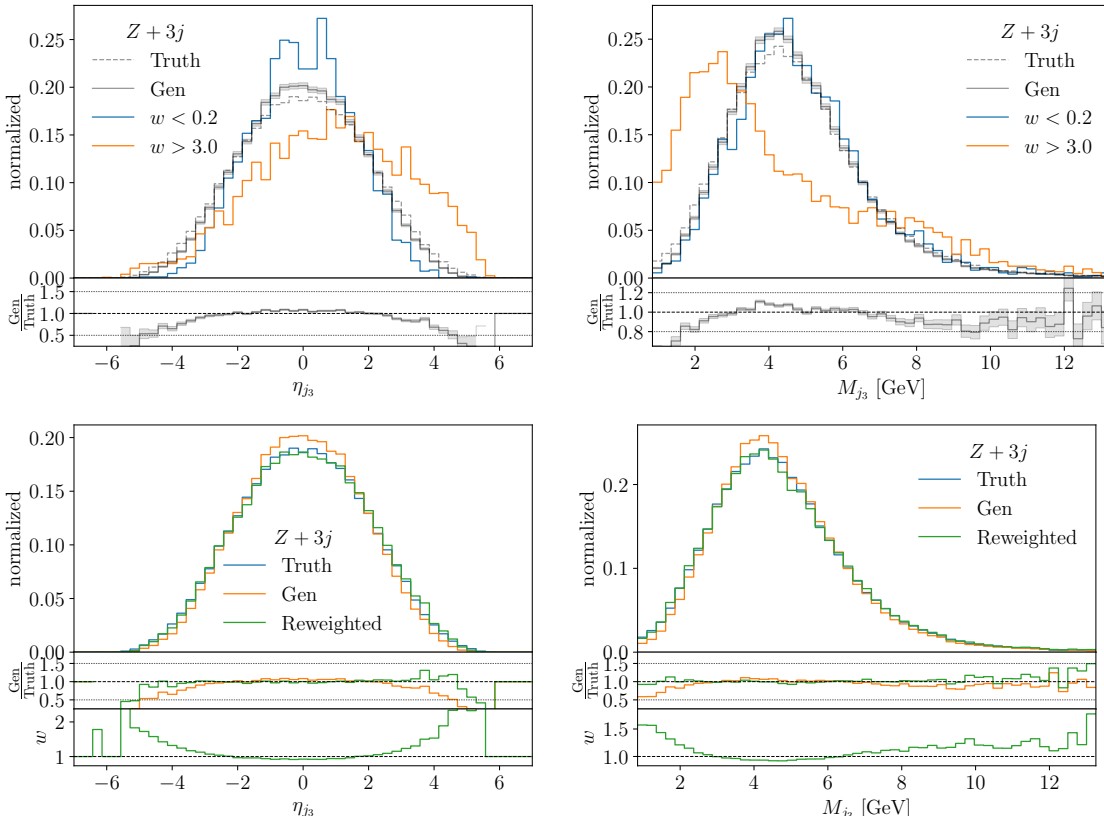

Figure 11: Critical kinematic distributions and for $Z + 3$ jets events from the state-of-the-art generator in different weight ranges (upper) and comparing generated data with truth (lower). The events with small weights are taken from the generated distribution, the events with large weights are taken from the truth distribution.

In Fig. 12 we first look at the correlation between the classifier weight $w(x)$ and the relative error $\sigma(p_{\text{model}})/\mu(p_{\text{model}})$ and the median of the Bayesian INN estimate as a function of $w$. For events with one and two jets there is a clear correlation, while for $Z + 3$ jet events missing features start to dominate the classifier, and the two uncertainty estimates lose their correlation. In the lower panels of Fig. 12 we show the pull distributions, normalizing the deviation of the generated from the true density (encoded in the classifier) by the uncertainty from the generative network. While we obtain a roughly Gaussian shape, its width is much smaller than we would expect. The reason for this is the problem of assigning an uncertainty to individual phase space points without taking their correlation into account.

We can understand the conservative uncertainty estimate of the Bayesian network from the kinematic observables. We use the same distribution of likelihoods as for reweighted distributions. Turning each distribution into a histogram taking the bin-wise means and standard deviations, we can also define an error bar for each histogram bin. In Fig. 13 we first show the same four distributions as in Fig. 10, but now with a Bayesian network uncertainty. For the smooth rapidity and momentum distributions the event reweighting only has a minor effect, corresponding to the observation that events with anomalous weights do not cluster in these distributions. The BNN uncertainty estimate is over-conservative in that it easily covers the deviation of the model from the truth and also the effect of event reweighting.

The situation changes for the $Z$-peak, where the network does well, the reweighting does not lead to a significant improvement of the sharp mass peak, but the uncertainty estimate there is too small. For $\Delta R_{jj}$ we see what happens if the (Bayesian) generative network ignores

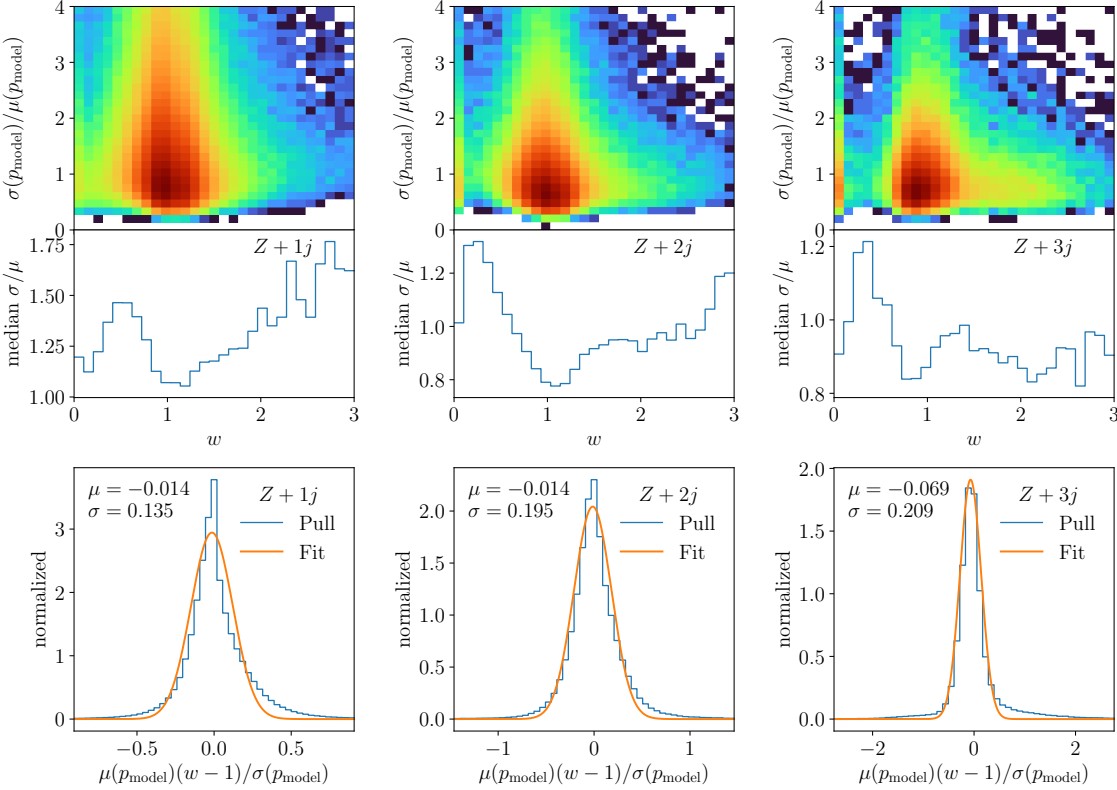

Figure 12: Top: correlation between the classifier weights and the relative standard deviation of the event weights from the Bayesian generator; Center: medians over the *w*-bins. Bottom: pulls combining the standard deviation of the event weight distribution with the error estimate from the Bayesian generator.

a feature altogether — in this case the missing collinear enhancement is not accounted for in density estimation and also not in the uncertainty estimation for the density. This suggests that the implicit bias of the generative networks does not allow it to capture the structure. On the other hand, the classifier identifies this failure mode, and the reweighted distribution reproduces the truth with high precision.

Finally, in the lower two panels of Fig. 13 we show a Bayesian uncertainty estimate for the challenging cases from Fig. 11. We already know that the classifier identifies the problematic phase space region correctly, and the reweighted events reproduce the truth distribution. The question is where this problem comes from. The Bayesian network output tells us if this problem is related to a lack of training data or to the network structure. We see that similar to the first two distributions the Bayesian uncertainty estimate easily covers the difference between generated events and truth, as well as the difference between generated and reweighted events. This clearly points towards a limitation in the training data, most likely just the size of the 3-jet dataset. As a side remark, Fig. 13 would not even have flagged these two distributions as problematic, this potentially crucial piece of information requires a dedicated study of events with anomalous weights.

## 6 Conclusions

Generative networks play an important role in the ML-transformation of LHC physics. They can be used for many tasks in event generation, simulation, and advanced analysis. This



Figure 13: Kinematic distributions for $Z + 3$ jets events from the Bayesian state-of-the-art generator, with bin-wise error bars on the generated events. The distribution include the set of Fig. 10 as well as the challenging distributions from Fig. 11.

comes with the requirement to control their precision in the density estimation over phase space systematically. A classifier is perfectly suited to control generative models, as motivated by established classifier reweighting. In addition to a single AUC value it provides us with a wealth of information on the strengths, weaknesses, and failure modes of the generative model.

We have applied a performance test based on the classifier weight distributions for three different generative tasks.[2] First, we have studied a not very realistic, but challenging modification of generated jet configurations, to find that these modifications can be identified and even corrected for by looking at jets with anomalous weights. Our second case were calorimeter showers, where the weight distributions identified types of showers that the generative model did not learn well, pointing us towards possible improvements of the generative setup. Finally, we have looked at an event generator for $Z$+jets events, for which the classifier weights again allow us to identify and understand the problems of the generative network training. For this case we also showed how our diagnostic can be embedded in a comprehensive precision and uncertainty framework for generative events.

Some standard failure modes appearing in our three applications and diagnosed by the weight distributions are: (i) missing features or missing tails in the generated events, leading to a tail of large weights $w \gg 1$ clustered in phase space; (ii) wrongly learned phase space boundaries or sharp cliffs, leading to a tail towards small weights $w \ll 1$, clustered in phase space; (iii) sharp features learned with reduced resolution, leading to a shift of peak of the weight distribution to values $w < 1$ and a compensating enhancement at finite $w > 1$, related to the amount of missing resolution and also clustered in phase space.

The clustering of anomalous weights in the interpretable phase space has, in all cases, allowed us to identify the physics reason behind the poorly performing generative network. Moreover, reweighting the events with the classifier weights over phase space allows us to improve the network and make sure that the weighted events do reproduce all key features.

Our study shows that a trained classifier can and should be used to analyze the performance of generative networks; the weight distributions not only tests the performance of the generator, it also allows us to identify failure modes, correct for shortcomings, and defines a key ingredient to the development of precision generators for particle physics.

## Acknowledgments

We would like to thank Anja Butter and Ramon Winterhalder for many useful discussions and for the close coordination with Ref. [68].

**Funding Information** CK and TP would like to thank the Baden-Württemberg-Stiftung for financing through the program *Internationale Spitzenforschung*, project *Uncertainties - Teaching AI its Limits* (BWST_IF2020-010). RD and DS are supported by the U.S. Department of Energy under Award Number DOE-SC0010008. TH is funded by the Carl-Zeiss-Stiftung through the project *Model-Based AI: Physical Models and Deep Learning for Imaging and Cancer Treatment*. This research is supported by the Deutsche Forschungsgemeinschaft (DFG, German Research Foundation) under grant 396021762 – TRR 257: *Particle Physics Phenomenology after the Higgs Discovery* and through Germany's Excellence Strategy EXC 2181/1 – 390900948 (the *Heidelberg STRUCTURES Excellence Cluster*).

## A  Classifier calibration

To gauge whether the classifiers used in our study have been well-trained (not overfitted, reasonably close to optimal), one important check is to inspect their calibration curves. The idea of the calibration curve is that a properly learned and optimal classifier $C(x)$ should return

---

[2]We collected the datasets used to train the classifiers in a Github repository which is publicly available at https://github.com/heidelberg-hepml/discriminator-metric.

the probability that $x$ is class 1, and $1-C(x)$ the probability that $x$ is class 0. Therefore, if we took all events $x$ in the training data (assumed to be balanced) for which $C(x) = C$, a fraction $C$ of them should be class 1. The differential way to write this is

$$\frac{\dfrac{dN_1}{dC}}{\dfrac{dN_1}{dC} + \dfrac{dN_0}{dC}} = C \,. \tag{A.1}$$

As in the main body of this paper, we will look at calibration curves in terms of the weights $w$. Using Eq.(1), we can turn Eq.(A.1) into a statement about the weights,

$$\frac{dN_1}{dw} = \frac{dN_0}{dw} w \,. \tag{A.2}$$

Equation (A.2) implies an equivalent way of plotting a calibration curve in weight space: divide the combined weight distribution in bins and calculate the ratio $N_{\text{truth}}/N_{\text{gen}}$ for each bin. According to Eq.(A.2), for a well-calibrated classifier these should agree. We show calibration curves, calculated following this method, for our classifiers in Fig. 14. We see that the classifiers are for the most part very well-calibrated. One possible exception is for $e^+$ at lower weights, but one should keep in mind this is one of the better generative models considered in this work (AUC=0.536), so nearly all the events are in the well-calibrated part of the calibration curve (with $w \approx 1$). Also, as we see in the discussion in Sec. 4.2 and in Fig. 17, even if the tails of the classifier are mis-calibrated, it can still extract poorly modeled regions of phase space and assign, if anything, too extreme weights to them. However, attention is needed when using them for reweighting.

To confirm that our calibration curves in Fig. 14 are indeed reasonably well-calibrated, we consider the case of an *overfitted* classifier obtained by training on two statistically identical GEANT samples. In Fig. 15 (left) we show the weight distribution for the test dataset after epochs 10 and 150. The middle panel shows the training and validation losses when training without learning rate scheduler. The right panel shows the calibration curve for epoch 150. The network learns to distinguish between the two GEANT samples and reweights one noise into the other. This guarantees that the weight distribution is symmetric around the maximum at $w = 1$. However, the weight distributions broaden during training. Because the difference between the two datasets is feature-less, this broadening is the same on the classifier training and test datasets. At the same time, Fig. 15 and Eq.(6) illustrate the benefit of studying the weight distributions: the AUC evaluated on the test dataset is stable at 0.5 during the entire training, but the weight distribution shows that the classifier is moving away from optimality. All in all, we see how all three diagnostics — weight distribution, training/validation losses, and calibration curve — indicate a poorly trained classifier, in stark contrast to the classifiers considered in this paper.

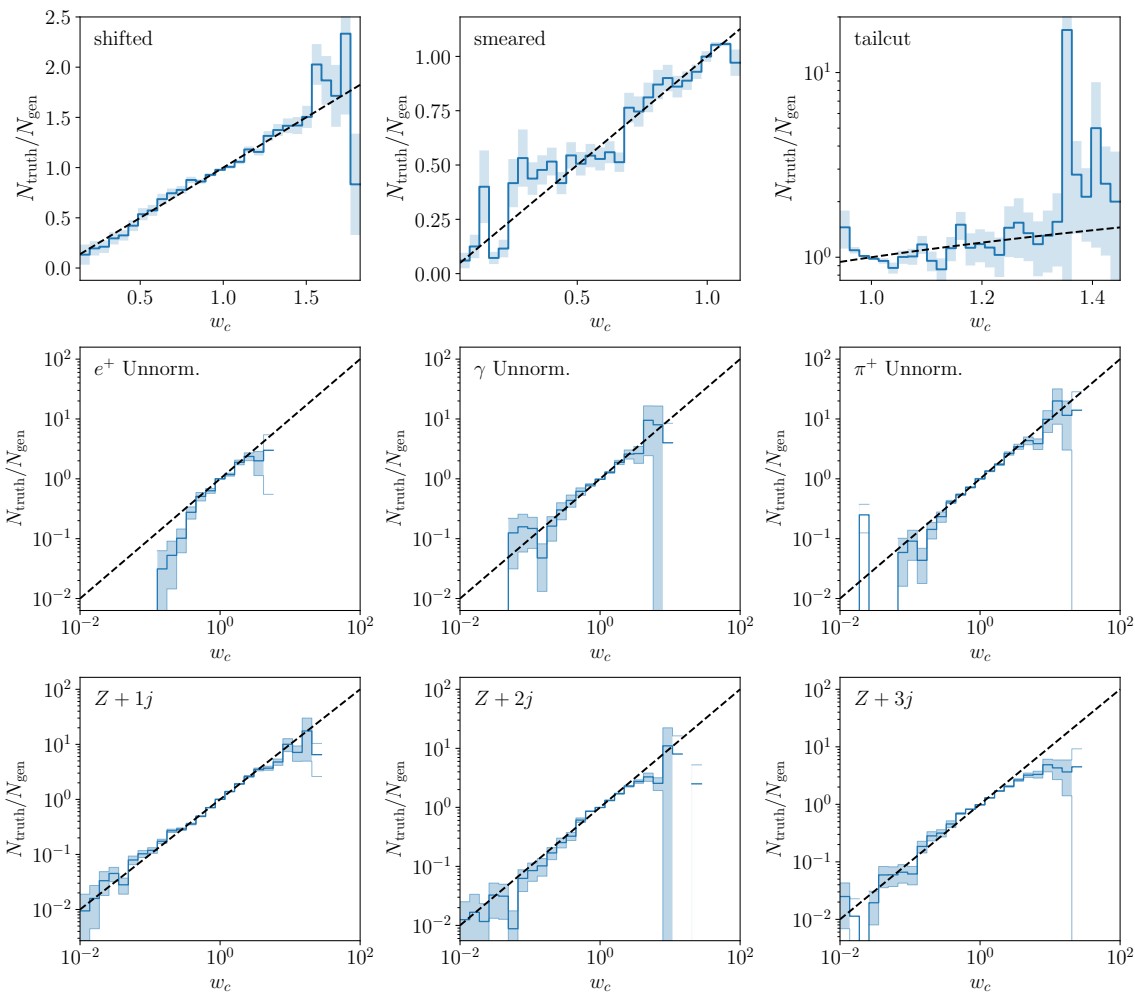

Figure 14: Calibration plots in weight space for the different discriminator models. Top to bottom: (i) jets with shifted, smeared and tailcut distortions; (ii) normalized showers for $e^+$, $\gamma$, and $\pi^+$; (iii) $Z + \{1, 2, 3\}$ jets using the state-of-the-art generator.

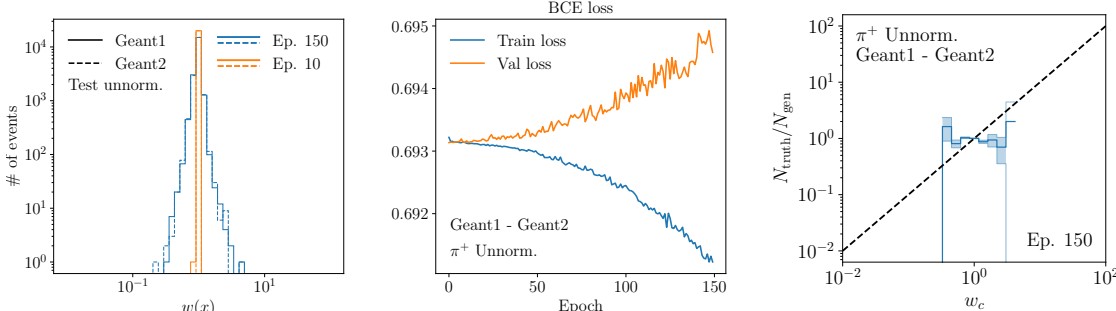

Figure 15: Left: weight distributions of test set for a classifier trained on two different GEANT samples for pion showers. Center: BCE loss function for training and validation. Right: calibration curve in weight space at epoch 150.

# B  Additional kinematic distributions

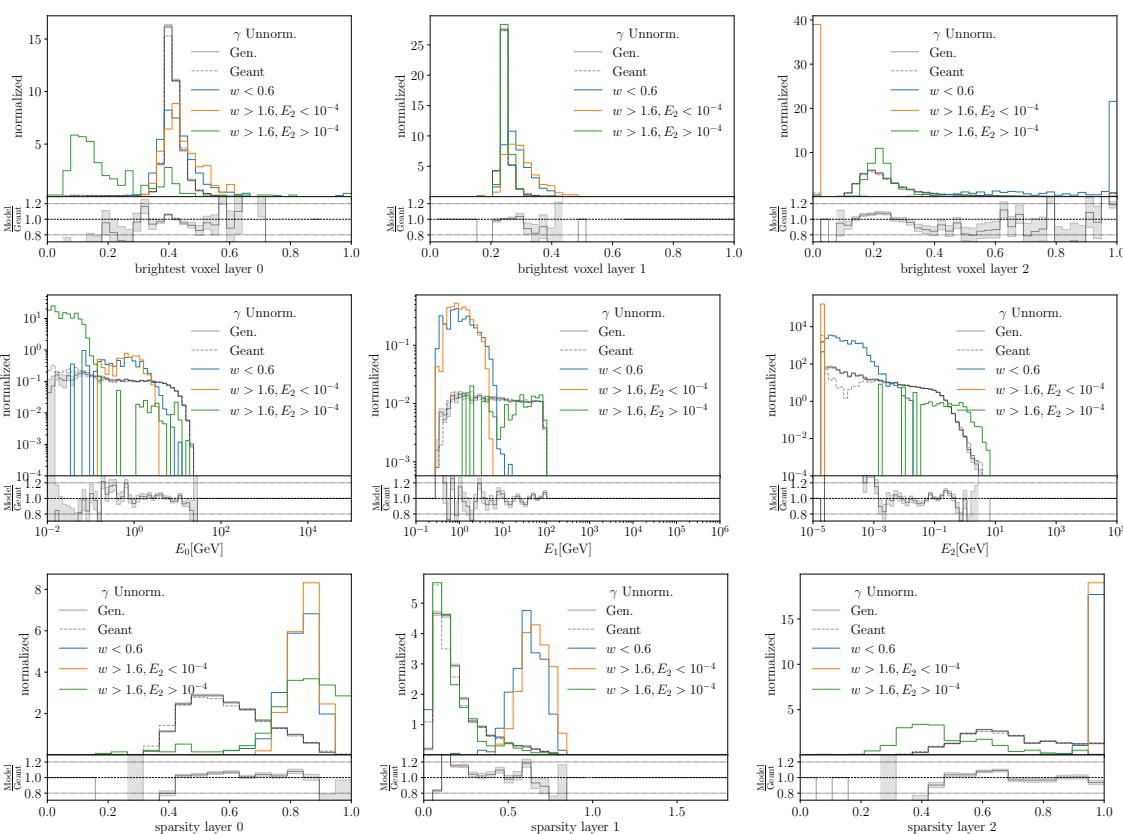

Figure 16: Clustering plots for $\gamma$: (i) reweighting is required at low energies, but the pattern is not just the energy; (ii) (orange) under-sampling of soft showers with zero energy deposition in layer-2; (iii) (blue) induced over-sampling of soft showers in layer-2; (iv) (green) under-sampling of delayed showers, low energy deposition in layer-0.

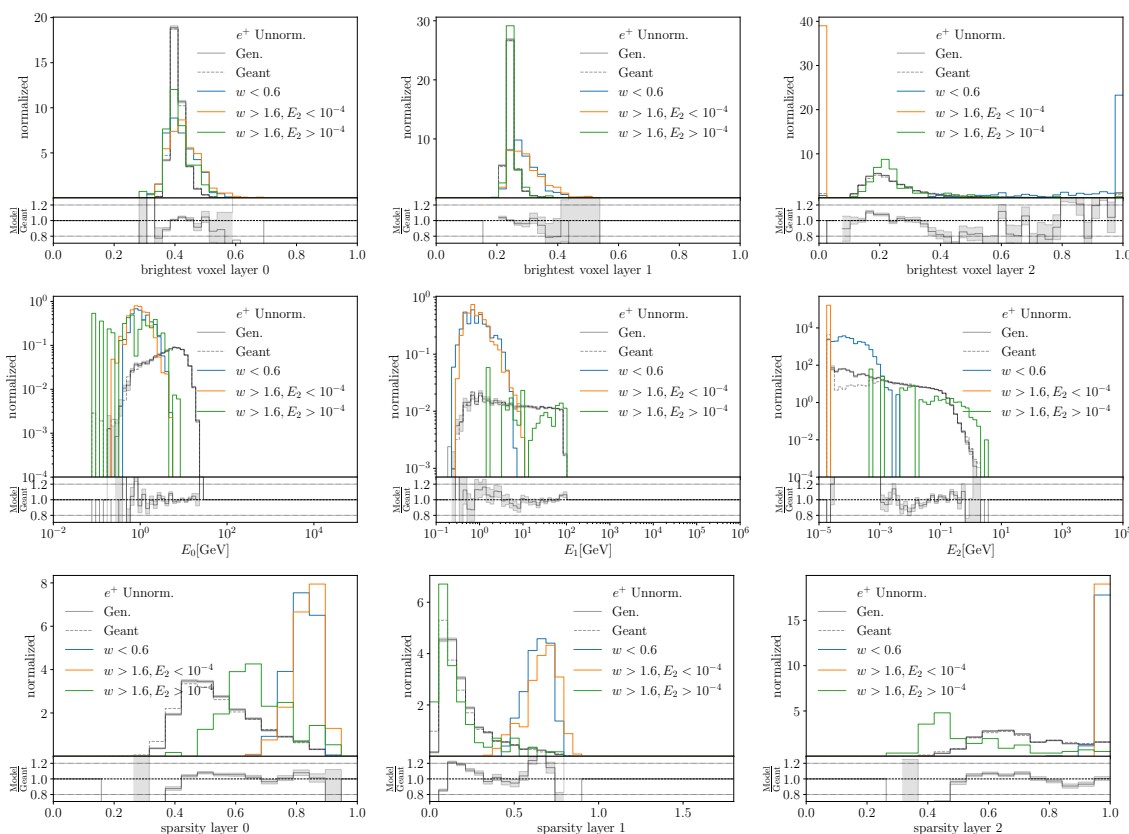

Figure 17: Clustering plots for $e^+$: similar pattern of $\gamma$ showers, expected given the similar physics and data structure.

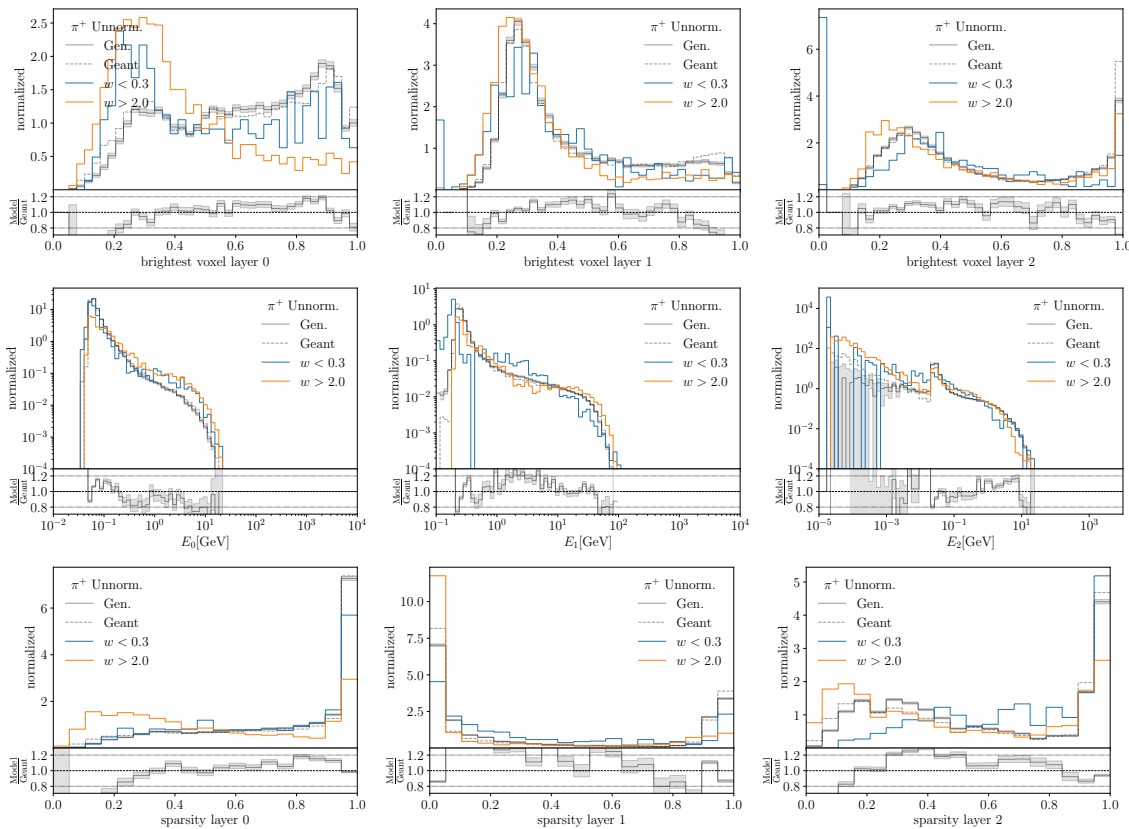

Figure 18: Clustering plots for $\pi^+$: (i) for the different energies the INN finds all features, but the balance between feature and continuum is not perfect; (ii) in both tails corrections at all energies are applied; (iii) the generator over-samples showers with no energy deposition in layer-1 and layer-2; (iv) large sparsity values are underestimated by the INN.

Figure 19: Set of kinematic distributions for $Z+3$ jets events from the state-of-the-art generator in different weight ranges. The events with small weights are taken from the generated distribution, the events with large weights are taken from the truth distribution.

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
