# Peer review of "How to Understand Limitations of Generative Networks"

_SciPost Physics, doi:SciPost Phys. 16, 031 (2024)_

## Round 1 · Referee Report · Xiangyang Ju (Referee 1) · 2023-8-17

Strengths
- demonstrate a new tool to understand the performance of generative models
- the tool can identify regions the generative model performs worse
- extensive studies on different tasks in particle physics
Weaknesses
- novelty is OK. The weighting distributions have been already used to examine the goodness of generative models by others, and classifiers have been used to improve generative models too. The idea presented here is not new.
- The impact is OK. Not clear to me how one would use the weight distributions to improve generative models except reweighting.
Report
However, the weakness of the paper is that it does not demonstrate how the identification of failure mode can be used to improve the generative model except reweighting, which is a method that has been exploited in many literatures to improve generated distributions from generative models. No strong evidence shows that using the weight distributions helps to “define a key ingredient to the development of precision generators for particle physics,” ingredients that can not be easily identified by examining the generated physics observables. For example, in p11, the authors discussed their findings in Figure 6. They identified several failure modes. However, how these identified ingredients can be translated to “key ingredients to the development of precision generators for particle physics” is not discussed nor demonstrated. How can one use that information to improve the model itself? Failure to establish the connections only means the weight distribution is a possible means for a posterior explanation of generated events.
Requested changes
p6 and p7, "we argue that the AUC is indeed the wrong metric." and "the AUC is basically 0.5." These are very misleading statements. An AUC of 0.495 is not "basically" 0.5. The classifier yielding an AUC of 0.495 renders orders of magnitude higher in background rejection for a low signal efficiency as shown in Figure 1 compared with an ideal classifier yielding an AUC of 0.5. Please rephrase both sentences.

---

## Round 1 · Referee Report · Anonymous (Referee 2) · 2023-9-22

Report
This paper investigates the use of ML-based classifiers to quantify the quality of ML-based generative models. The classifier output is transformed into a likelihood ratio estimator that is studied to explore the properties of shortcomings in the learned generative model. Given the growing interest in deep generative models, there is a need for new tools to quantify their performance. This paper is thus timely. I also find the studies to be serious and the examples highly relevant. SciPost is a good venue for this work.
Before I can recommend publication, I have some comments and suggestions:
-
The key assumption underlying this work is that classifiers are more accurate estimators of likelihood ratios than generative models. The authors argue this on empirical grounds based on their previous works, but I think it would be helpful to make reference to the ML literature on discriminative versus generative classifiers. The latter are not universally worse, and are often better when the training datasets are small. That is usually not a relevant case for HEP, but it is worth a comment.
-
The main methodology is exactly what underlies a GAN. A classifier ("discriminator") is trained and used to provide feedback to improve the generator. It might make sense to make this connection explicitly. Unlike the GAN training, you are interrogating the full distribution and not just the impact on the loss.
-
A related question is if / how you can best use the classifier diagnostic information to improve the generative model. You talk about reweighing, which does improve the performance, but not of the generative model directly and may have some undesirable consequences that are not discussed (e.g. dilution of statistical power). In addition to the GAN setup, would you please comment on how one might use this diagnostic information to improve the generator itself?
-
I was somewhat surprised to not find any clustering analysis other than by hand looking at particular distributions. Have you looked at where high and low w events cluster using any standard clustering method (in the original space or in a transformed space a la TSNE or similar?)
-
Please consider making your datasets and software public! (if I were in charge of SciPost, I would make this a requirement for publication).

---

## Round 2 · Author Response

We thank the referees for their feedback on the manuscript.

A major point raised by both reports is the application of classifier knowledge to improve the generative network. Our studies cover examples where the generators are well understood and, therefore, we can easily pin down failure modes from the weight distributions. In general this is not true, the classifier can highlight biases in the generative model which are not visible from histograms of 1-d observables. This knowledge can directly be used during training, we include a discussion of different methods in the manuscript, or it can be used a posteriori as a diagnostic tool to highlight failure modes, for instance in the tail of the weight distribution. We demonstrate its effectiveness in the paper by matching features of the weight distribution to failure modes in the high-level features. Improving the generator afterwards is not the point of our paper, addressing the identified issues has to be done in a problem-dependent way, for instance by changing the input parameterization or the network architecture. Generally, the workflow would still follow our discussion: - look for tails or structure in the weight distriutions; - match these failure modes to physics aspects of the data and/or failures in the generator with manual or automated clustering methods. We added a simpified example of possible improvement direction in section 4.2 but we leave the details for future work.

---

## Round 2 · List of Changes

Reply to Report 2:

1) We added a comment on the ML literature on discriminative versus generative classifiers.

2-3) We added a discussion of the similarities and differences of using a classifier for reweighting, in the training of a GAN, for directly improving the generative network and as a diagnostic tool.

4) We believe that an analysis based on projection in a lower dimensional space, e.g. PCA or TSNE, is not more interpretable than studying physically motivated observables, especially for problems where complex features are known. For example the sharp $\Delta$R cut for event generation.

5) We included the Github repository in the manuscript which includes links to the datasets used to train the classifiers.

Reply to Report 1:

Requested change:
Fig. 1 is an example where a large correction found by the classifier is visible in the ROC curve (as you describe, background rejection is orders of magnitude higher), but it has almost no impact on the numerical value of the AUC, which is close to 0.5. Therefore, it confirms our statement that the AUC (i.e. the single number, not the full ROC curve) as a performance metric is insensitive to this failure mode. We rephrased the second statement to make the distinction between the sensitivity of the AUC and the ROC curve more clear.

---

## Editorial Decision

published